

# Higher serum β2-microglobulin is a predictive biomarker for cognitive impairment in spinal cord injury

Zhonghao Cui[1], Shuai Wang[2], Yanke Hao[3] and Yuanzhen Chen[1]

[1] Shandong First Medical University & Shandong Academy of Medical Sciences, Bone Biomechanics Engineering Laboratory of Shandong Province, Shandong Medicinal Biotechnology Center (School of Biomedical Sciences), Neck-Shoulder and Lumbocrural Pain Hospital of Shandong First Medical University, Jinan, Shandong Province, China
[2] Shandong University of TCM, Jinan, Shandong Province, China
[3] Orthopedics Department, The Affiliated Hospital of Shandong University of TCM, Jinan, Shandong Province, China

## ABSTRACT

**Objective:** Recent studies have suggested that high levels of β2-microglobulin are linked to cognitive deterioration; however, it is unclear how this connects to spinal cord injury (SCI). This study sought to determine whether there was any association between cognitive decline and serum β2-microglobulin levels in patients with SCI.
**Methods:** A total of 96 patients with SCI and 56 healthy volunteers were enrolled as study participants. At the time of enrollment, specific baseline data including age, gender, triglycerides (TG), low-density lipoprotein (LDL), systolic blood pressure (SBP), diastolic blood pressure (DBP), fasting blood glucose (FBG), smoking, and alcohol use were recorded. Each participant was assessed by a qualified physician using the Montreal cognitive assessment (MoCA) scale. Serum β2-microglobulin levels were measured using an enzyme-linked immunosorbent assay (ELISA) reagent for β2-microglobulin.
**Results:** A total of 152 participants were enrolled, with 56 in the control group and 96 in the SCI group. There were no significant baseline data differences between the two groups ($p > 0.05$). The control group had a MoCA score of $27.4 \pm 1.1$ and the SCI group had a score of $24.3 \pm 1.5$, with the difference being significant ($p < 0.05$). The serum ELISA results revealed that the levels of β2-microglobulin in the SCI group were considerably higher ($p < 0.05$) than those in the control group ($2.08 \pm 0.17$ g/mL compared to $1.57 \pm 0.11$ g/mL). The serum β2-microglobulin level was used to categorize the patients with SCI into four groups. As serum β2-microglobulin levels increased, the MoCA score reduced ($p < 0.05$). After adjustment of baseline data, further regression analysis showed that serum β2-microglobulin level remained an independent risk factor for post-SCI cognitive impairment.
**Conclusions:** Patients with SCI had higher serum levels of β2-microglobulin, which may be a biomarker for cognitive decline following SCI.

Corresponding author
Yuanzhen Chen,
chenyuanzhen2011@163.com

## INTRODUCTION

Spinal cord injury (SCI), a neurological condition with a high rate of disability, has a significant negative impact on sufferers, their families, and society as a whole. According to the National Center for SCI Statistics, there are 12,500 new cases of SCI in North America each year and more than 90% of these cases are traumatic and frequently affect male adults (*McDonald & Sadowsky, 2002*; *Simpson et al., 2012*). SCI is a severe neuropathological condition that is underlined by spinal cord ischemia, inflammatory response, oxidative stress, and neuronal death. Although numerous treatment approaches have been proposed and some researchers have dedicated their careers to creating medications that support neuroprotection and nerve regeneration, the therapeutic benefit is still elusive (*Alizadeh, Dyck & Karimi-Abdolrezaee, 2019*; *Young, 1993*; *Shinozaki et al., 2021*; *Shi et al., 2021*). Therefore, it is crucial to identify targets that are connected to neuroprotection, immune modulation, and nerve regeneration in SCI.

β2-Microglobulin, a 17 kD protein, constitutes the light chain of the MHC I molecule (*Floege & Ketteler, 2001*; *Loureiro & Faísca, 2020*). It can fold into the usual seven-chain immunoglobulin fold and is the active component of the adaptive immune system. β2-Microglobulin can be found in several cells, produced and expelled by lymphocytes, and found in the blood of healthy individuals (*Minguet et al., 2007*; *Cai et al., 2022*). β2-Microglobulin and the MHC I molecule in the brain, independent of their conventional immune activity, can control normal brain development, synaptic plasticity, and behavior (*Ohta et al., 2011*; *Paschen, Melero & Ribas, 2022*; *Wu et al., 2021*).

We hypothesized that β2-microglobulin is involved in SCI-related cognitive impairment in light of the recent reports linking β2-microglobulin to cognitive impairment, the correlation between systemic β2-microglobulin levels and cognitive decline, and the identification of β2-microglobulin as a potential pro-dementia factor associated with neurological damage. To establish a theoretical foundation for determining whether β2-microglobulin is involved in cognitive impairment following SCI, this study measured the cognitive level and blood β2-microglobulin level in patients with SCI and examined any potential relationships between the two.

## METHODS

### Participants

Patients with SCI who visited the Neck-Shoulder and Lumbocrural Pain Hospital between July 2020 and June 2022 were continually screened. The inclusion criteria were as follows: (1) presenting to the facility within 48 h of acute SCI onset; (II) age 18–80; (III) no surgical treatment; and (IV) no referral to another institution. The exclusion criteria were as follows: (I) receiving surgical care or being referred for surgery; (II) being 18 or 80 years old; (III) the presence of a severe systemic disease or tumor; (IV) having a history of an acute infection or having surgery for trauma within 2 weeks before the start of the study; (V) having prior cognitive impairment; and (VI) patients or their relatives not agreeing to participate in the study. Additionally, we enrolled regular folks as controls. The flowchart of the study is shown in Fig. 1. All participants or members of their families completed

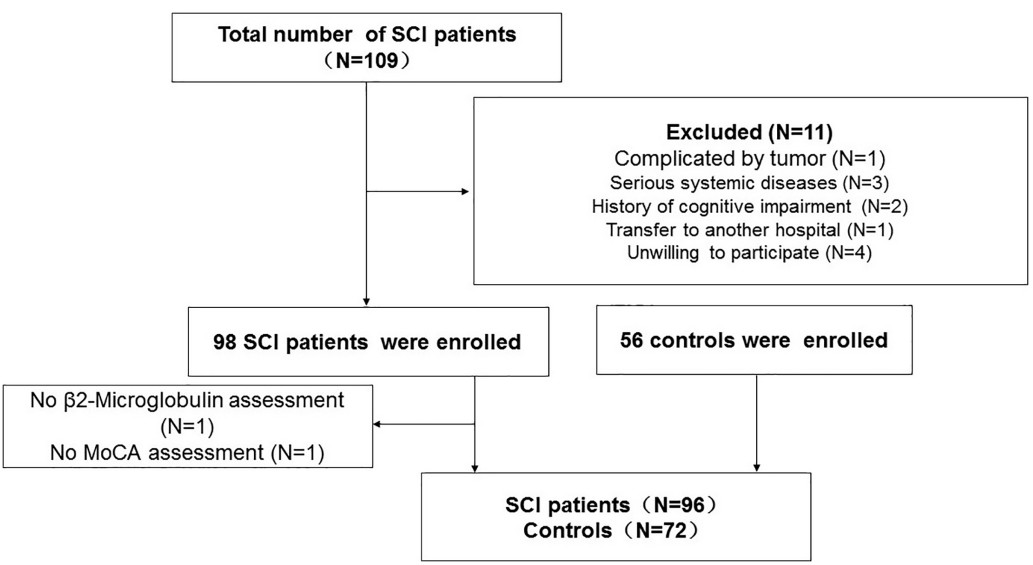

**Figure 1 Flowchart of the study.** SCI, Spinal cord injury; MoCA, Montreal cognitive test.

consent forms indicating their assent to take part in the study. The study was approved by the Ethics Committee of the Neck-Shoulder and Lumbocrural Pain Hospital (Approved ID: 202004).

## Clinical baseline data

Trained staff members gathered clinical baseline data at the time of enrolment. Patients and their known carers answered a questionnaire that we provided. Age, gender, triglycerides (TG), low-density lipoprotein (LDL), fasting blood glucose (FBG), systolic blood pressure (SBP), and diastolic blood pressure (DBP) were among the clinical baseline data. Investigators were blinded to the baseline data.

## Cognitive screening

A cognitive evaluation of all patients was performed using the Montreal cognitive assessment (MoCA) scale. The cognitive screening was completed within 48 h of enrollment. The MoCA was established by Ziad Nasreddine in Montreal, Canada in 1996. The test has since gained widespread use in detecting cognitive impairment. The test includes a 30-point total score and a 10-min detection time. Short-term memory, attention, concentration, and executable performance were among the cognitive domains that were examined. With a sensitivity of 90% for detecting moderate cognitive impairment (MCI), substantially greater than mini-mental state examination (MMSE), MoCA was demonstrated in 2000 to be a highly sensitive instrument for the early detection of MCI (*Wang et al., 2020*).

## Serum $\beta$2-microglobulin level

All participants had their peripheral blood drawn within 24 h of enrollment. A volume of 5 mL of fasting blood was drawn from the patients in the morning, stored at room

**Table 1  Clinical baseline data for participants.**

|  | Controls (*n* = 56) | SCI (*n* = 96) | *p* |
|---|---|---|---|
| Age, years | 54.6 ± 7.3 | 54.2 ± 6.7 | 0.732 |
| Gender, male/female | 41/15 | 76/20 | 0.40 |
| TG, mmol/L | 1.47 ± 0.18 | 1.51 ± 0.20 | 0.219 |
| LDL, mmol/L | 2.35 ± 0.24 | 2.39 ± 0.26 | 0.348 |
| SBP, mmHg | 117.1 ± 10.3 | 116.2 ± 11.4 | 0.628 |
| DBP, mmHg | 76.4 ± 6.7 | 77.0 ± 7.5 | 0.622 |
| FBG, mmol/L | 5.1 ± 0.9 | 5.4 ± 1.1 | 0.086 |
| Smoking, n (%) | 14 | 25 | 0.887 |
| Drinking, n (%) | 19 | 37 | 0.570 |
| MoCA, (scores) | 27.4 ± 1.1 | 24.3 ± 1.5 | <0.001 |
| β2-Microglobulin, μg/ml | 1.57 ± 0.11 | 2.08 ± 0.17 | <0.001 |

**Note:**
SCI, Spinal cord injury; TG, triglycerides; LDL, low-density lipoprotein cholesterol; SBP, systolic blood pressure; DBP, diastolic blood pressure; FBG, Fasting blood-glucose; MoCA, Montreal cognitive test.

temperature for 30 min, and centrifuged at 12,000 g for 15 min. The serum was divided into smaller amounts and kept in a −80 °C freezer. β2-Microglobulin reagent (R&D, Minneapolis, MN, USA) was used to measure the serum levels of β2-microglobulin. Please consult the prior literature and relevant product instructions for the precise ELISA operating specification (*Xu et al., 2021*).

## Statistical analysis

All statistical analyses were performed using SPSS 22.0. Quantitative data were presented as continuous variables, normally distributed data were shown as (n ± SD), and count data were presented as n or rate. T-tests or analyses of variance were used to compare the two groups. The association between β2-microglobulin and MoCA score was evaluated using p for trend. It was possible to predict risk factors for cognitive impairment using multivariate regression analysis. The cutoff value for statistical significance for all statistics was set at 0.05.

# RESULTS

## Clinical baseline data

Age, gender, TG, LDL, FBG, SBP, DBP, and FBG were among the clinical baseline data used in this investigation. Table 1 summarizes these clinical baseline data, and the findings revealed that there was no appreciable difference between the SCI and control groups ($p > 0.05$).

## MoCA score and serum $\beta$2-Microglobulin level

The control group had a MoCA score of 27.4 ± 1.1, whereas the SCI group had a score of (24.3 ± 1.5), with the difference being significant ($p < 0.05$). The serum ELISA revealed that the levels of β2-microglobulin in the SCI group were considerably higher ($p < 0.05$) than

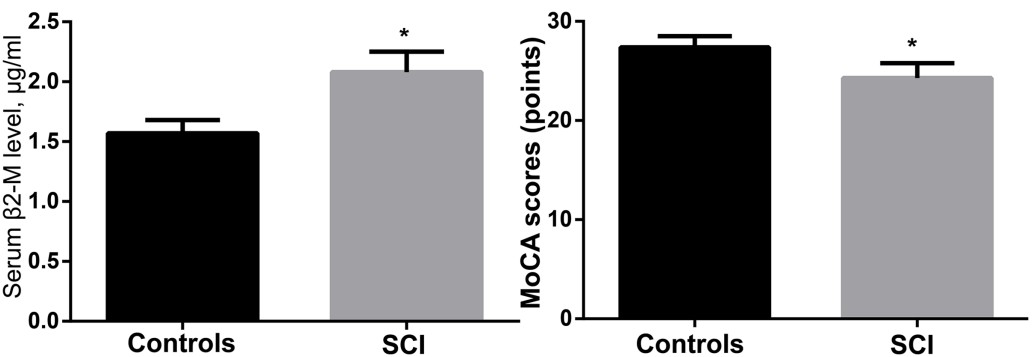

**Figure 2 Comparison of MoCA scores and serum β2-microglobulin levels between the two groups.**
*$p < 0.05$.             

**Table 2 Correlation between serum β2-microglobulin levels and cognitive impairment.**

| Variable | Serum β2-Microglobulin levels (ug/ml) | | | | $p$ |
|---|---|---|---|---|---|
| | Q1 | Q2 | Q3 | Q4 | |
| MoCA score | 25.1 ± 1.8 | 24.7 ± 1.6 | 24.1 ± 1.4 | 23.3 ± 1.2 | <0.001 |

Note:
  MoCA, Montreal cognitive test.

those in the control group (2.08 ± 0.17 g/mL compared to 1.57 ± 0.11 g/mL). Table 1 and Fig. 2 show the findings of the MoCA test and serum β2-microglobulin levels.

### Correlation between MoCA score and serum $\beta$2-microglobulin level

The link between the MoCA score and the serum β2-microglobulin level was evaluated using the p for trend test. The findings demonstrated a significant downward trend in the MoCA score with increasing blood β2-microglobulin levels from Q1 to Q4 ($p < 0.05$), indicating a possible inverse relationship between the two (Table 2).

### Multiple regression analysis

A crucial technique for adjusting for risk factors and identifying potential causal links is multiple regression analysis. In our investigation, various models were constructed. Age and gender were adjusted for in Model 1. Smoking and alcohol use were further accounted for in Model 2. Also, TG, LDL, DBP, SBP, and FBG were included in Model 3. The results of the three models (Table 3) all indicated that β2-microglobulin was an independent risk factor for cognitive impairment following SCI ($p < 0.05$).

## DISCUSSION

The study found a considerable increase in the blood level of 2-microglobulin following SCI. This increase was strongly inversely linked with the MoCA score. After correcting for the usual risk factors for cognitive impairment, further investigation showed that β2-microglobulin still has the potential to function as a biomarker for predicting cognitive impairment after SCI.

**Table 3 Regression analysis of serum β2-microglobulin levels and MoCA scores.**

|  | MoCA scores | |
| --- | --- | --- |
|  | Regression coefficient | *p* values |
| Model 1 | 0.352 | <0.001 |
| Model 2 | 0.271 | <0.001 |
| Model 3 | 0.218 | 0.047 |

Note:
Model 1: adjusted for age and gender; Model 2: further adjusted for smoking and drinking; Model 3: further adjusted for SBP, DBP, FBG and serum β2-Microglobulin levels. MoCA, Montreal cognitive test; SBP: systolic blood pressure; DBP: diastolic blood pressure; FBG, fasting blood-glucose.

It is believed that SCI and cognitive decline are related (*Sachdeva et al., 2018*; *Craig et al., 2017*; *Davidoff, Roth & Richards, 1992*). One of the main factors contributing to cognitive impairment following SCI is thought to be cardiovascular disease. Most patients with SCI experience orthostatic hypotension and autonomic dysreflexia, which are characterized by risky changes in systemic blood pressure. According to available data, prolonged hypotension and hypertension in healthy adults can seriously deteriorate cerebrovascular health and impair cognitive function. As a result, managing blood pressure instability is critical, and doing so may help lessen the effects of cognitive impairment following SCI (*Sachdeva, Nightingale & Krassioukov, 2019*). Additionally, studies have demonstrated a relationship between systemic blood pressure and blood flow velocity and cognitive impairment following SCI (*Wecht et al., 2018*). Our earlier research has demonstrated that neurogenesis, oxidative stress, and inflammation contribute to cognitive impairment following SCI (*Liu et al., 2022*; *Sun et al., 2021*; *Chen et al., 2020*).

MHC class I molecules are made up of the proteins α1, α2, α3, and β2-microglobulin and are found in all nucleated cells. These molecules are encoded by the B2M gene. By obstructing the middle cerebral artery, the Xuzhou Medical University research team established a rat model of focal cerebral ischemia. They discovered that the level of β2M in the brain tissue, serum, and cerebrospinal fluid of the rats with acute cerebral infarction was significantly elevated; however, the levels gradually decreased during the recovery period. The infarct volume and cognitive impairment can be decreased by using RNA interference to inhibit the expression of β2M in the acute phase of stroke. The mechanism may be related to the inhibition of glial cells, caspase-1 and NLRP3 inflammasome activation, and downstream inflammatory pathways (*Chen et al., 2023*). A research team at Central South University investigated the impact of toll-like receptor 4 (TLR4) on age-related cognitive deterioration caused by β2-microglobulin. The findings revealed that β2-microglobulin-induced age-related cognitive decline occurs *via* the TLR4 signaling pathway as WT mice exogenously injected with β2-microglobulin exhibited age-related cognitive decline, increased TLR4 mRNA expression, and increased levels of interleukin (IL)-1, (TNF)-α, and hippocampal apoptotic neuronal death. A key neuroprotective method for preventing age-related cognitive decline is presented in this study (*Zhong et al., 2020*). Researchers from the University of South China, led by Tang Xiaoqing, discovered that $H_2S$ can reduce the effects of β2-microglobulin-induced cognitive dysfunction by

restoring blocked autophagic flux in the hippocampus. This finding raises the possibility that H$_2$S functions as a defense against β2-microglobulin-induced cognitive dysfunction (*Chen et al., 2019*). Italian researchers also discovered that the development of Alzheimer's disease is associated with an increase in serum β2-microglobulin, although additional clinical research is required to validate this finding (*Dominici et al., 2018*).
The aforementioned findings all point to the involvement of β2-microglobulin in the etiology of cognitive impairment.

This study, which is a useful addition to the mechanism of β2-microglobulin-induced cognitive impairment, established for the first time that β2-microglobulin is implicated in cognitive impairment following SCI. However, the study also has some drawbacks. First, we used a cross-sectional design with a limited sample size, which requires future validation. Second, we did not follow up and are unaware of the dynamic relationship between β2-microglobulin and cognitive function. No cell or animal research was performed to investigate the mechanism.

## CONCLUSIONS

Greater serum levels of β2-microglobulin are a biomarker that predicts cognitive impairment after SCI. Serum β2-microglobulin and a thorough analysis of the causes of cognitive decline after SCI may shed fresh light on the rehabilitation of SCI patients and enhance their quality of life.

### Funding
The authors received no funding for this work.

### Competing Interests
The authors declare that they have no competing interests.

### Author Contributions
- Zhonghao Cui conceived and designed the experiments, performed the experiments, analyzed the data, prepared figures and/or tables, authored or reviewed drafts of the article, and approved the final draft.
- Shuai Wang conceived and designed the experiments, performed the experiments, prepared figures and/or tables, authored or reviewed drafts of the article, and approved the final draft.
- Yanke Hao conceived and designed the experiments, performed the experiments, prepared figures and/or tables, authored or reviewed drafts of the article, and approved the final draft.
- Yuanzhen Chen conceived and designed the experiments, performed the experiments, analyzed the data, prepared figures and/or tables, and approved the final draft.

## Human Ethics

The following information was supplied relating to ethical approvals (*i.e.*, approving body and any reference numbers):

The Ethics Committee of Neck-Shoulder and Lumbocrural Pain Hospital of Shandong First Medical University gave its approval to our investigation (202004).

## Field Study Permissions

The following information was supplied relating to field study approvals (*i.e.*, approving body and any reference numbers):

Field experiments were approved by the Research Council of Neck-Shoulder and Lumbocrural Pain Hospital of Shandong First Medical University (202004).

## Data Availability

The raw measurements are available in the Supplemental File.

## Supplemental Information

Supplemental information for this article can be found online at http://dx.doi.org/10.7717/peerj.15372#supplemental-information.

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
