# Peer review of "Higher serum β2-microglobulin is a predictive biomarker for cognitive impairment in spinal cord injury"

_PeerJ, doi:10.7717/peerj.15372_

## Round 0.1 · original submission · Major Revisions

Please answer the Reviewers' Comments, carefully.

·

Basic reporting

This manuscript is clearly structured and generally conveys the authors' view that elevated serum β2-microglobulin is a predictive biomarker of cognitive dysfunction after spinal cord injury.
However, it should be noted that serum β2-microglobulin is expressed in many cognitive disorders, such as Alzheimer's disease, and that serum β2-microglobulin does not appear to be specific for cognitive impairment after spinal cord injury. In addition, there are some issues in the manuscript that need to be carefully revised by the authors for better reading.
The English language should be improved to ensure that an international audience can clearly understand your text. I suggest you have a colleague who is proficient in English and familiar with the subject matter review your manuscript, or contact a professional editing service.

Experimental design

The study methodology was too simple and flawed, only testing for differential expression of serum β2-microglobulin levels and MoCA scores in the control population and in patients with spinal cord injury. The MoCA scale was completed within 48 hours for subjects in the manuscript, implying that cognitive impairment due to spinal cord injury was present within 48 hours for patients in this study? How do the authors explain this phenomenon. It seems unacceptable that patients with spinal cord injury that may have resulted in cognitive impairment after more than 48 hours would appear to have been affected under this study protocol. What is the association between TG, LDL, FBG, SBP, DBP and FBG indicators in the clinical baseline data of the study, and serum β2-microglobulin? If not, what is the significance of testing these indicators? Simply to derive the correlation between serum β2-microglobulin and cognitive impairment in the final regression analysis?

Validity of the findings

There is no doubt that the authors' team seems to have conducted some preliminary studies on cognitive dysfunction in spinal cord injury, possibly involving the involvement of factors such as oxidative stress and inflammation. However, the authors in this manuscript do not discuss in depth through which mechanism serum β2-microglobulin contributes to cognitive dysfunction after spinal cord injury or what are the causes of elevated serum β2-microglobulin due to spinal cord injury? A manuscript should contain the author's insights into the content of the study and an in-depth discussion of the reasons why. Therefore, it is recommended that the author focus on the above mentioned elements for better understanding by the reader.

Additional comments

It is recommended that authors should provide a brief description of the current status, epidemiology, and characteristic presentation of cognitive impairment associated with spinal cord injury patients to add to the completeness of the article.

Reviewer 2 ·

Basic reporting

Cognitive impairment after SCI is a common clinical complication, affecting patients' quality of life and prognosis. This manuscript compared baseline data, cognitive scales, and the levels of beta 2-Microglobulin in SCI patients and normal controls, and the results suggested that beta 2-Microglobulin might be a potential biomarker, which has certain clinical implications.In general, the design of this paper is reasonable and ethical. It is a simple case-control study.

Experimental design

The design of this article is basically reasonable, with no obvious flaws. But there are some problems:
1.The inclusion and exclusion criteria are vague. Please redescribe them.
2.Beta-2-microglobulin levels are associated with renal function, so why didn't baseline data include indicators of renal function?
3.Why wasn't cerebrospinal fluid beta 2-Microglobulin detected?
4.The language needs polishing.

Validity of the findings

1.The authors need more centers and populations to further test their conclusions.
2.In vitro and in vivo experiments are necessary to explore the underlying mechanisms.

Additional comments

Some abbreviations should be spelled in full when they first appear. There are two "3.2" in the results part.

Reviewer 3 ·

Basic reporting

no comment

Experimental design

the inclusion and exclusion criteria of patients in the two groups were not rigorous.

Validity of the findings

Statistical Analysis (t test)was not correct to analysis four group results.

Additional comments

This study sought to determine whether there was any connection between cognitive decline and serum ³2-Microglobulin levels in SCI patients. the authors dound that patients with SCI had higher serum levels of ³2-Microglobulin. however, this sutdy had some mayor flaw. The inclusion and exclusion criteria of patients in the two groups were not rigorous. there was not enough evidence to support the relationship between cognitive decline and SCI patients. the result was simple and the discussion need to improve.

---

## Round 0.2 · accepted · Accept

The third reviewer has not responded to my invitation to re-review the manuscript. After carefully checking, I consider that the authors have responded to all reviewer's questions very well.

·

Basic reporting

First, the authors have carefully corrected the grammatical errors in the article and have professionally touched up the article, which will help the article to be read by a wider range of physicians.

Experimental design

Secondly, the authors have answered some questions about the study protocol and ethics, and the authors have acknowledged the limitations of the study protocol, but I do not think it affects the acceptance and publication of this manuscript.

Validity of the findings

The data on which the conclusions are based must be provided or made available in an acceptable discipline-specific repository. The data should be robust, statistically sound, and controlled.

Additional comments

Finally, I hope that the author's team will carry out further research in this area

Reviewer 2 ·

Basic reporting

No comment.

Experimental design

No comment.

Validity of the findings

No comment.